# Investigation of Ancient Architectural Painting from the Taidong Tomb in the Western Qing Tombs, Hebei, China

**Peng Fu [1], Ge-Le Teri [1], Jing Li [1], Jia-Xin Li [1], Yu-Hu Li [1,\*] and Hong Yang [2,\*]**

[1] Engineering Research Center of Historical Cultural Heritage Conservation, Ministry of Education, School of Materials Science and Engineering, Shaanxi Normal University, No. 620, West Chang'an Avenue, Chang'an District, Xi'an 710119, China; fupeng@snnu.edu.cn (P.F.); terigelesnnu@163.com (G.-L.T.); lijing9669@126.com (J.L.); tuoyuanxinggungun@snnu.edu (J.-X.L.)

[2] The Imperial Palace Museum, Beijing 100009, China

\* Correspondence: liyuhu@snnu.edu.cn (Y.-H.L.); yanghong960518@163.com (H.Y.)

**Abstract:** The Taidong Tomb in the imperial tombs of the Qing dynasties has great aesthetic value and a rich history. In this study, we conducted the first investigation ever performed on the raw materials used in the paintings in the Taidong Tomb. Energy dispersive X-ray spectroscopy (EDX), polarized light microscopy (PLM), X-ray diffraction (XRD), micro-Raman spectroscopy (m-RS), Fourier-transform infrared spectroscopy (FTIR) and pyrolysis–gas chromatography–mass spectrometry (Py-GC/MS) were used to comprehensively analyze the painting of Long'en Hall, Xipei Hall and the ceiling of Minglou. In the conclusion of the study, the paintings were found to contain natural mineral and synthetic pigments, including atacamite ($Cu_2Cl(OH)_3$), azurite ($2CuCO_3 \cdot Cu(OH)_2$), vermilion (HgS), carbon black (C), anglesite ($PbSO_4$), white lead ($2PbCO_3 \cdot Pb(OH)_2$), synthetic emerald green ($Cu(CH_3COO)_2 \cdot 3Cu(AsO_2)_2$) and ultramarine (($Na,Ca)_8(AlSiO_4)_6(SO_4,S,Cl)_2$). This allows us to conclude that some of the architectural paintings were repainted in the mid-to-late 19th century. The mortar layer may consist of brick ash (albite, gismondine), lime water, tung oil and flour. The fiber layer material may be ramie. Researching the raw materials of the paintings in the Taidong Tomb is of great value because it provides scientific data for the future preservation of the paintings in the tomb.

**Keywords:** Taidong Tomb; pigment; mortar; ancient architectural painting

## 1. Introduction

The Western Qing Tombs are located in Yi County, Hebei Province, China. These royal tombs of the Qing Dynasty are of important historical, artistic and scientific value. They were protected as a national key culture protection unit, and later the Western Qing Tombs were also added as a world cultural legacy site by the 24th World Heritage Committee in November 2000. UNESCO maintains and conserves this area, which is a popular tourist attraction.

The Western Qing Tombs are among the largest royal tombs of the Qing Dynasty. Taidong Tomb is located in the Western Qing Tombs. It is the tomb of Empress Xiaoshengxian—mother of Emperor Qian Long. It is positioned in Dongzheng Valley, about 1500 m northeast of Emperor's Yongzheng's tomb. Taidong Tomb is the largest and most complete of the empresses' tombs. It is a masterpiece of architectural beauty with typical Qing dynasty marks, which include Long'en Hall, Xipei Hall, Dongpei Hall, Shenku Hall, Minglou Hall. The Taidong Tomb has cultural significance and historical value for study on Qing dynasty funerary architecture techniques.

The Qing dynasty attached great importance to the construction and placement of the tombs because they believed in the continuation of life after death. The tombs are surrounded by mountains

next to a river. This deals with the principles of Chinese geomancy or Feng shui, which is arranging objects so that good energy flows through one's environment. The tombs are exquisite and superior constructions that reflect the highest technical level of tomb building of the Qing dynasty. Stones are precisely cut, carved marble archways supported by marble pillars, In the Qingxi tombs, there are many color paintings, especially in the inner eaves that were produced during the middle and late Qing dynasty, including the Qianlong period. These paintings are extremely rare and well-preserved. They represent the work of the greatest masters of art during this period and are an indispensable part of the Qingxi tombs. They are fully authentic, and the integrity reflects their value. Therefore, researching the pigments, painting materials and techniques of the color paintings of the Qingxi tombs has profound importance for the preservation of the culture of the Qing dynasty. A panoramic photo of Taidong Tomb is shown in Figure 1.

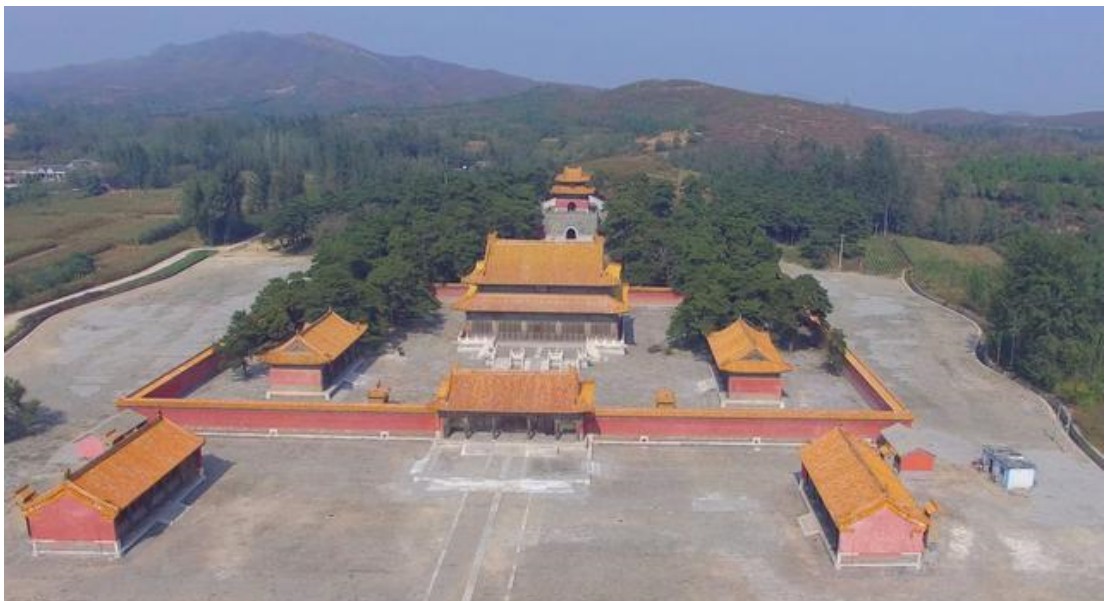

**Figure 1.** Panoramic photos of Taidong Tombs.

To date, the raw materials used for the Taidong paintings have never been studied. To begin exploring the materials used, the structure thickness of the sample and pigments particle were analyzed by a polarized light microscopy. Next, we used the energy dispersive X-ray detector (EDX) [1,2], micro-Raman spectroscopy (m-RS) [3–6] and X-ray diffractometer (XRD) [7,8] to perform a multianalytic study of the pigment. Finally, we used the XRD [9] to study the inorganic filler materials in the mortar. To analyze the binding materials in the mortar, the instrument Fourier-transform infrared spectroscopy (FTIR) and the pyrolysis gas chromatography mass spectrometry (Py-GC/MS) method [10] were used.

## 2. Materials and Experiment

### 2.1. Sample Information

The samples—which include green, white, blue, red and black pigments—were collected from the damaged places of Long'en Hall, Xipei Hall and the ceiling of Minglou in the Taidong Tomb. As shown in Table 1 and Figure 2, the sample numbers of the ceiling of Minglou are labeled as M1, M2, M3, and M4; the sample numbers of Long'en Hall are labeled as L1, L2, and L3; and the sample numbers of Xipei Hall are labeled as X1, X2, and X3.

**Table 1.** Long'en Hall, Xipei Hall, ceiling of Minglou sampling location.

| Number | Sample Positon |
|---|---|
| M1<br>M2<br>M3<br>M4 | Ceiling of Minglou |
| X1<br>X2<br>X3 | Xipei Hall |
| L1<br>L2<br>L3 | Long'en Hall |

**Figure 2.** Micro pictures of pigment samples.

## 2.2. Sample Preparation

### 2.2.1. Cross-Section Preparation

Samples from Long'en Hall, Xipei Hall and the ceiling of Minglou were selected to make the cross section. First, we poured half of the resin (transparent cold mount; Laizhou Weiyi Experimental Machinery Manufacture co. ltd., Laizhou, China) into the mold. We let it cool and solidify at room temperature. Then, we put the X3, L2 samples into the mold with tweezers and filled the mold with the resin. We used a low-density cutting machine to obtain the cross section, and finally used 4000–12,000 mesh sandpaper to polish the cross sections.

### 2.2.2. Fiber Preparation

The fibers in the mortar were removed with tweezers and put into ultra-pure water. It took 10 h to disperse the fibers segment into single fibers using ultrasonic vibration at a temperature of 60 °C. The fiber was then placed on the glass slide with tweezers and covered with a round cover glass ($\varphi$ 12 mm). Herzberg stain was dropped along the edge of the cover glass, so that Herzberg stain was immersed in the whole cover glass. Herzberg stain is an iodide–iodine solution mixed with a zinc chloride solution. The Herzberg stain used in this work was a mixture of zinc chloride solution (50 g $ZnCl_2$ dissolved in 25 mL distilled water) and iodine solution (0.25 g $I_2$ and 5.25 g KI dissolved in 12.5 mL distilled water) we used a polarizing microscope to observe the fiber [11].

### 2.2.3. Preparation of Pigment Sample

Elbow tweezers were used to scrape the pigments from the sample surface. Part of these powder samples were analyzed by X-ray diffractometer (EDX-7000, Shimadzu, Kyoto, Japan) or micro-Raman spectroscopy (m-RS, in Via Reflex, Renishaw, Wotton-under-Edge, UK) The remaining pigment particles were dispersed in ethanol and then added to the glass slide with a dropper. A round cover glass ($\varphi$ 12 mm) was used to cover the pigment particles. The resin (MeltmountTM, refractive index = 1.662) was heated to 60–70 °C. After the resin liquified, it was dripped along the edge of the cover glass to immerse it completely. Finally, the crystal properties of pigment particles were observed under polarized light microscopy (PLM; Olympus BX53M; Shinjuku, Tokyo, Japan).

### 2.2.4. Preparation of Mortar Sample

The mortar of sample L2 was ground with agate mortar and examined by the X-ray diffractometer and FTIR. The sample, which was approximately 50 μg, was placed directly in a pyrolysis instrument with a mass fraction of 3 μL of 20% tetramethylammonium hydroxide solution (TMAH). After soaking it for 1 h, the sample was then put into an automatic sampler instrument and the instrument started to crack the sample at 600 °C. The cracked product was then analyzed by gas chromatogram-mass spectrometry. Finally, the isolated compounds were identified using NIST14 and NIST14 s mass spectrometry databases.

## 2.3. Instrument

Energy dispersive X-ray spectroscopy (EDX-7000, Shimadzu, Kyoto, Japan) was used mostly to identify the elements of a pigment; it was equipped with high performance silicon drift detector and X-ray tube (Rh target); test range: 11NA–92U; collimator: 1, 3, 5, 10 mm $\varphi$.

Polarized light microscopy was used to observe the cross section of the sample. The instrument was equipped with an achromatic polarized light module acquisition lens, 5–20× objective lens to get detailed information for cross section of the sample. The crystal characteristics of pigment particles were observed under horizontal and perpendicular polarized light.

Micro-Raman spectroscopy was used to determine the composition of the pigments. Scanning range: 100–4000 $cm^{-1}$; white and green pigments used a laser wavelength of 785 nm. The laser

wavelength of black, red and blue pigment was 532 nm. The exposure time: 30 s; accumulations: 1–3; laser power: 1–2 mW.

A X-ray diffractometer was used to analyze the composition of pigments and inorganic fillers in the mortar equipped with Cu Kα radiation ($k$ = 1.54059 Å). The maximum output power was 9 kW; Tube voltage: 45 kV; Tube current: 200 mA; 2θ range: 10°–90°; scanning step: 0.01°.

Fourier-transform infrared spectroscopy (FTIR; Vertex70; Bruker, Billerica, MA, USA) was used to study the composition of binding materials in the mortar. Spectral range: 350–5000 cm$^{-1}$.

Pyrolysis-gas chromatography-mass spectrometry (Py-GC/MS) was composed of a pyrolysis instrument (EGA/PY-3030D, Frontier Labs, Koriyama, Japan) and a gas chromatography mass spectrometry (GC/MS; QP2010Ultra, Shimadzu, Kyoto, Japan). Pyrolysis instrument parameters: thermal cracking temperature 600 °C; thermal cracking time 10 s; syringe temperature 250 °C; the interface temperature of syringe and chromatograph 320 °C.

Gas chromatography mass spectrometry parameters: chromatography column SLB-5MS (5% diphenyl/95% dimethyl siloxane). The initial temperature of the oven where the column was located was 50 °C, and it was maintained for 5 min. Then it was increased at 3 °C/min to 292 °C for 3 min. The carrier gas helium was used. The pressure before column was 15.4 kPa with a flow rate of 0.6-mL/min and a separation rate of 1:100.

Ionization voltage of mass spectrometer: 70 eV; scanning 0.5 s, mass–charge ratio (*M/Z*) of 50 to 750.

## 3. Results and Discussion

### *3.1. Cross-Section*

The cross-section of the sample was observed by the resin embedding procedure. One sample was selected for cross section analysis in each area of Long'en Hall (L2) and Xipei Hall (X3).

According to the cross-section in Table 2, the samples of L2 and X3 were three-layered structures. The L2 sample had three main layers: layer "a" was made of pigment; layer "b" was made of mortar; layer "c" was made of fiber. The green pigment, layer "a", was about 66 μm. The mortar layer "b" was about 969 μm, and the fiber layer "c" was about 1.2 mm. X3 also had a three-layer structure. The green pigment, layer "a", was about 32 μm. The mortar layer "b" was about 2.2 mm, and the fiber layer "c" was more than 1 mm. The thickness of mortar layer of L2 and X3 samples was different. According to the results of pigment analysis, the possible reason was that the L2 green pigment sample was repainted in the mid-19th century, resulting in different mortar thickness of the two samples.

### *3.2. Pigments*

#### 3.2.1. Green Pigment

The green pigment EDX test in Table 3 of L2 and X1 samples were comprised mostly of the elements Cu and As. The Raman spectra of L2 and X1 are shown in Figure 3e,g. The Raman bands at 122, 154, 175, 217, 243, 294, 325, 371, 429, 492, 539, 951, 1355, and 1441 cm$^{-1}$, corresponded to emerald green $Cu(CH_3COO)_2 \cdot 3Cu(AsO_2)_2$. According to the Raman spectrum, there was a large number of bands in the 100–400 cm$^{-1}$ that was mainly attributed to the vibration of Cu–O and As–O [12]. The 950 and 1440 cm$^{-1}$ bands correspond to the stretching vibrations of acetate groups C–C and –CO$_2$, which were the two key bands that distinguished copper arsenate mineral ($Cu_5(AsO_4)_2(OH)_4$) and Scheele's green ($Cu(AsO_2)_2$) [13]. Emerald green was confirmed as green pigments of L2 and X1. This was an imported pigment, which was invented and used in the early 19th century Europe [14] and was introduced to China in the middle of the 19th century [15]. Since the Taidong Tomb was built during the reign of Yongzheng in the mid-18th century, this fact indicates that the lower part of the Xipei Hall and the outer eaves of the Long'en Hall were repainted during the latter part of the Qing dynasty. As shown in Table 3, the green pigments used in the M1 and X3 samples contain Cu and Cl, which could be atacamite [16]. In addition, X3 samples contain Pb element, which may be mixed with other

pigments. The micro-Raman spectroscopy analysis could not determine the green samples of M1 and X3, which may be due to the strong fluorescent background caused by pollutants in the samples; further analysis by XRD is shown in Figure 4a,b. According to JCPDS, it was determined that M1 and X3 were mainly made of atacamite ($Cu(OH)_3Cl$; 71–2027). More substances were found in the X3 sample other than $Cu(OH)_3Cl$. According to the JCPDS, the corresponding substance was $PbSO_4$, indicating that X3 green pigment was doped with white pigment anglesite ($PbSO_4$; 72–1854) [17]. Under horizontal polarized light in Figure 5a, the sample (atacamite) green pigment particles appear dark green. Because there was no obvious shape, it was characteristic of the natural mineral pigment. Under perpendicular polarized light in Figure 5b, the crystal particles were yellowish-green. Under horizontal polarized light in Figure 5e, the diameter of the green pigment particles in L2 sample (emerald green) was about 20–50 µm and was round or fan-shaped, indicating that it was a synthetic pigment. The pigment particles appear darker in the middle and brighter at the edges. Under perpendicular polarized light in Figure 5f, the particles exhibit a green center and light blue periphery.

**Table 2.** Cross sections of L2 and X3 samples (dotted red line is used to divide the cross-sections of the layers).

| Sample Number | Cross-Section Micrograph | Serial Number | Composition | Thickness | Total Number of Layers |
|---|---|---|---|---|---|
| X3 |  | a | Green pigment | 66 µm | 3 |
| | | b | Mortar | 969 µm | |
| | | c | Fiber | 1200 µm | |
| L2 |  | a | Green pigment | 32 µm | 3 |
| | | b | Mortar | 2200 µm | |
| | | c | Fiber | >1000 µm | |

**Table 3.** Elemental composition of the samples.

| Sample Number | Main Elements of EDX Spectrum (wt.%) | | | | |
|---|---|---|---|---|---|
| M1 | Cu (62.91) | Cl (13.82) | Si (12.11) | Ca (4.73) | K (2.16) |
| M2 | Cu (58.13) | Ca (20.04) | Si (18.37) | S (3.37) | Fe (2.21) |
| M3 | Hg (47.52) | S (36.73) | Si (7.42) | K (4.04) | Ca (2.80) |
| M4 | Pb (64.85) | Ca (19.09) | Si (7.43) | S (6.30) | Fe (1.11) |
| X1 | Cu (23.81) | As (20.84) | Si (16.36) | Ca (14.57) | S (8.45) |
| X2 | Ca (20.60) | Si (20.03) | Al (16.38) | K (15.68) | S (11.04) |
| X3 | Cu (42.91) | Cl (15.82) | Pb (14.11) | Si (10.73) | Ca (10.16) |
| L1 | Ca (31.71) | Si (29.01) | S (19.7) | K (4.55) | Fe (4.51) |
| L2 | Cu (36.95) | As (36.01) | Si (8.06) | Ba (7.53) | S (3.03) |
| L3 | Si (27.99) | Ca (26.13) | Al (20.88) | S (14.89) | K (6.47) |

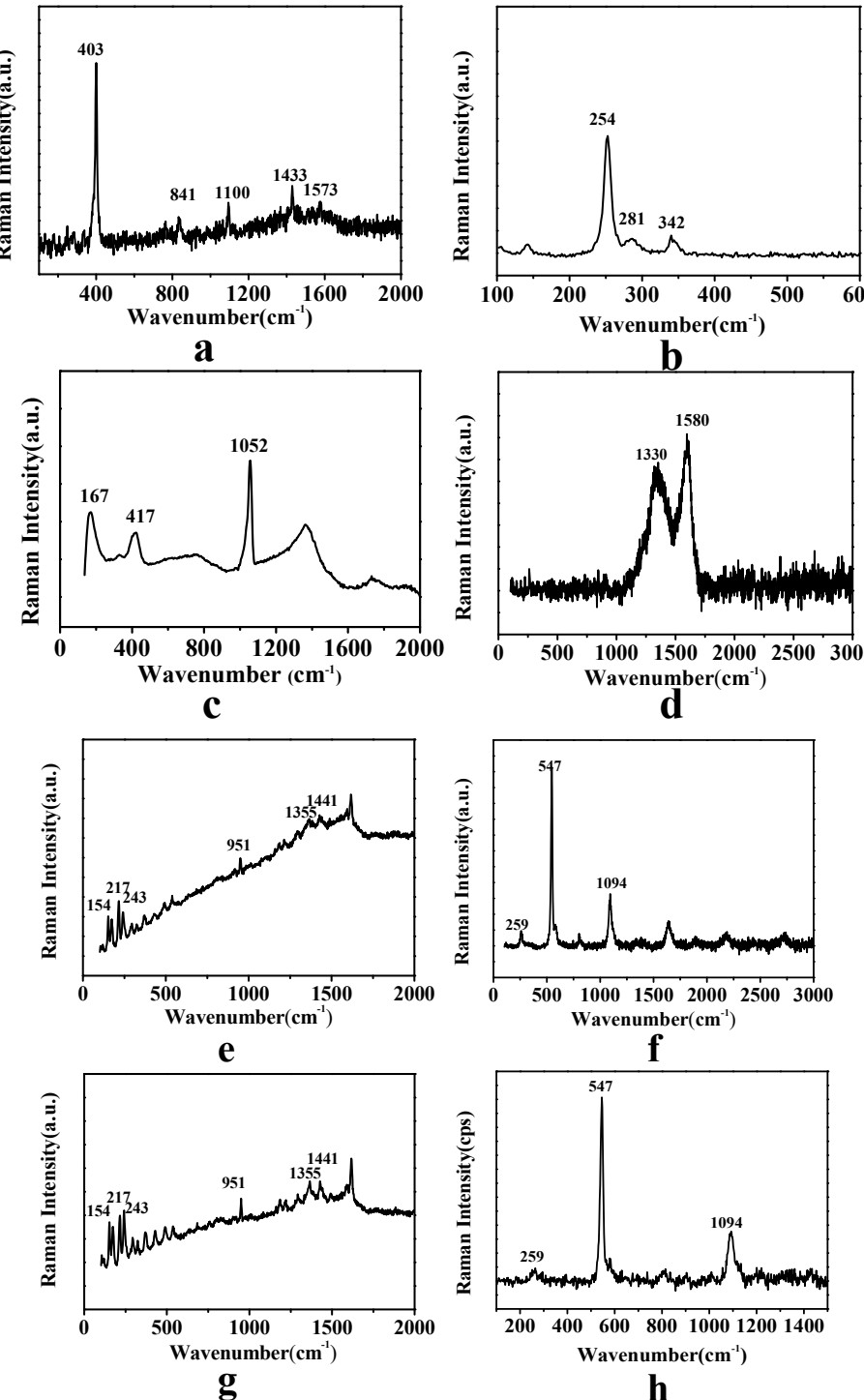

**Figure 3.** (**a**) Raman spectra for the blue pigment (azurite; $2CuCO_3 \cdot Cu(OH)_2$) in M2 sample; (**b**) Raman spectra for the red pigment (vermilion; HgS) in M3 sample; (**c**) Raman spectra for the white pigment(lead-white; $2PbCO_3 \cdot Pb(OH)_2$) in M4 sample; (**d**) Raman spectra for the black pigment (carbon black; C) in L1 sample; (**e**) Raman spectra for the green pigment (emerald green; $Cu(CH_3COO)_2 \cdot 3Cu(AsO_2)_2$) in L2 sample; (**f**) Raman spectra for the blue pigment (ultramarine; $(Na,Ca)_8(AlSiO_4)_6(SO_4,S,Cl)_2$) in L3 sample; (**g**) Raman spectra for the green pigment (emerald green; $Cu(CH_3COO)_2 \cdot 3Cu(AsO_2)_2$) in X1 sample; (**h**) Raman spectra for the blue pigment (ultramarine; $(Na,Ca)_8(AlSiO_4)_6(SO_4,S,Cl)_2$) in X2 sample.

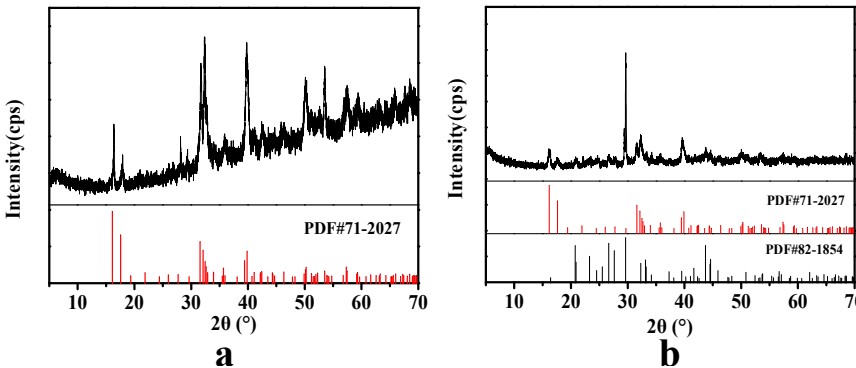

**Figure 4.** (**a**) XRD patterns for the green pigment (atacamite; $Cu(OH)_3Cl$) in M1 sample; (**b**) XRD patterns for the green pigment(atacamite and anglesite; $Cu(OH)_3Cl$ and $PbSO_4$) in X3 sample.

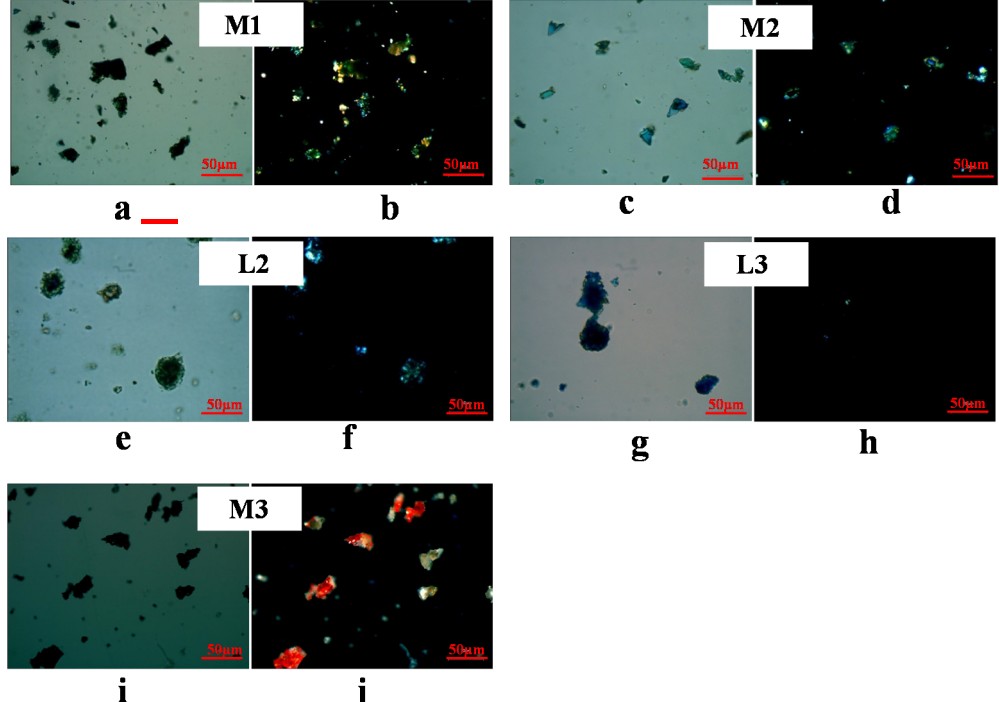

**Figure 5.** Pictures (**a,c,e,g,i**) are horizontal polarized light. Pictures (**b,d,f,h,j**) are perpendicular polarized light. M1: polarized light microscopy (PLM) image of atacamite; M2: PLM image of azurite; L2: PLM image of emerald green; L3: PLM image of ultramarine; M3: PLM image of cinnabar.

### 3.2.2. Blue Pigment

The main elements of blue pigment in sample M2 were Cu, Ca, Si, Pb and Fe determined by EDX in Table 3. To date, the common blue pigment containing Cu may be azurite. The main components of blue pigments in L3 and X2 were Ca, Si, Al, K and S elements in Table 3, which may be ultramarine [8]. For further verification, we used micro-Raman spectroscopy analysis. The Raman results of the blue pigment layer of sample M2 in Figure 3a. The Raman bands were at 403, 841, 1100, 1433, and 1573 cm$^{-1}$, which corresponds to azurite [$2CuCO_3 \cdot Cu(OH)_2$] [18]. For azurite pigment, the bands found in the spectral interval 0–600 cm$^{-1}$, were attributed to the Cu–O band vibration. the bands found in the spectral interval the 600–1400 cm$^{-1}$ belongs to the carbonate group vibration [19]. At near 1095 cm$^{-1}$ band was caused by the symmetric $\nu_1$ $CO_3^{2-}$ vibration of azurite [20]. The Raman result of blue pigments from L3 and X2 are shown in Figure 3f,h. The main peak position was near 259, 547, and 1094 cm$^{-1}$, whose corresponding substance was ultramarine, $(Na,Ca)_8(AlSiO_4)_6(SO_4,S,Cl)_2$ [21].

The near 259 cm$^{-1}$ band was the $\nu_2$ bending vibration of S$^{3-}$ ions and near 549 cm$^{-1}$ bands was $\nu_1$, a symmetric stretching vibration of S$^{3-}$ ions [19]. Under horizontal polarized light in Figure 5c, The pigment particles of M2 sample (azurite) were smaller in the range of 10–25 μm, the pigment particles were pale blue; however, some of them were green, which may be malachite. Azurite and malachite were associated minerals [22]. Therefore, green pigment (malachite) could be observed in azurite of the sample M2 (Figure 2), and green pigment (malachite) could also be observed in azurite (Figure 5). The pigment particles were fracture rock forms and belong to natural mineral pigments. Under horizontal polarized light in Figure 5g, the size of pigment (ultramarine) particles in sample L3 was not uniform, the pigment particles were dark blue and the crystal was round, indicating that it was a synthetic pigment and had complete extinction under perpendicular polarized light in Figure 5h.

### 3.2.3. Red Pigment

The EDX in Table 3 shows that M3 red pigment mainly contained Hg and S elements, which is speculated to be vermilion. In Figure 3b, vermilion was based on the three bands of Raman spectrum, 254 cm$^{-1}$ was the most obvious vibration, followed by two smaller bands in 281 and 342 cm$^{-1}$, which corresponds to vermilion (HgS) [6]. The results were consistent with those of EDX. The M3 sample (vermilion) pigment particles were in long strip or fracture rock form EDX in Table 3 shows that M3 red pigment mainly contained Hg and S elements, which was speculated to be vermilion. In Figure 3b, the assumption of vermilion was based on the three bands of Raman spectrum, 254 cm$^{-1}$ was the most obvious vibration, followed by two smaller bands in 281 and 342 cm$^{-1}$, which corresponds to vermilion (HgS) [6]. The results were consistent with those of EDX. M3 sample (vermilion) pigment particles were in long strip or fracture rock form in Figure 5i, with a fiery red and yellowish color under perpendicular polarized light in Figure 5j.

### 3.2.4. Black Pigment

Because EDX was limited by testing (only 11NA-92U could be tested), it did not provide important information about the element, but it could also exclude the presence of lead dioxide and iron black. As shown in Figure 3c, the D peak was near 1330 cm$^{-1}$, and the G peak was near 1580 cm$^{-1}$ of black pigment were consistent with characteristic peak of carbon black for the L1 black pigment [21]. Carbon black generally comes from incomplete combustion of wood and bones. If carbon black was formed by burning organic bones, there were obvious bands at 960 cm$^{-1}$ [21,23]; however, the black pigment Raman spectrum of L1 sample showed no characteristic bands in the near of 960 cm$^{-1}$. This indicates that the sample was not carbon black formed by organic bones; we presume that the black carbon originated from the burning of wood.

### 3.2.5. White Pigment

Through EDX in Table 3, white pigment contained a large number of Pb elements. It was speculated that the white pigment may be a mineral pigment containing lead. The samples of M4 were analyzed by micro-Raman. There were two noticeable Raman peaks at 417 and 1052 cm$^{-1}$, which corresponds to reported characteristic Raman peaks of lead-white (2PbCO$_3$·Pb(OH)$_2$). The peak at 1052 cm$^{-1}$ was the complete symmetric vibration of $\nu_1$, and at peak around 165, 417 cm$^{-1}$ were the Pb–O band vibration [19].

Finally, through the preliminary investigation of the three areas of Long'en Hall, Xipei Hall and ceiling of Minglou, the composition of the architectural painting was determined, as shown in Table 4.

### 3.3. Mortar and Fiber

During the Ming and Qing Dynasties, the production process of the mortar in architectural paintings was very complicated. According to the observation of cross section in Figure 2, the architectural paintings of Taidong Tomb was similar to the process of Yi Ma Wu Hui of the Qing Dynasty [9]. The materials that were mainly used include tung oil, flour, lime water and fiber [24].

We analyzed the mortar layer b and the fiber layer c of the L2 samples in Figure 2. The type of fiber was determined by Herzberg stain, and the mortar layer b of L2 was ground into powder by agate mortar and characterized by X-ray diffraction and FTIR.

**Table 4.** Analysis results of pigment from Taidong Tomb.

| Sample Number | Main Elements of EDX Spectrum | Composition |
| --- | --- | --- |
| Ceiling of Minglou (M1) | green pigment | atacamite |
| Ceiling of Minglou (M2) | blue pigment | azurite |
| Ceiling of Minglou (M3) | red pigment | vermilion |
| Ceiling of Minglou (M4) | white pigment | lead white |
| Xipei Hall (X1) | green pigment | Paris green |
| Xipei Hall (X2) | blue pigment | ultramarine |
| Xipei Hall (X3) | green pigment | Atacamite + anglesite |
| Long'en Hall (L1) | black pigment | carbon black |
| Long'en Hall (L2) | green pigment | Paris green |
| Long'en Hall (L3) | blue pigment | ultramarine |

First, we analyzed the fibers in the mortar using Herzberg stain. The fibers are viewed under a microscope before treatment, and it was clear that the fibers were composed of multiple single fibers as shown in Figure 6a. Then, the fiber was boiled with ultra-pure water for 10 min. Three percent hydrogen peroxide was used for ultrasonic vibration in ultra-pure water at 60 °C for 3 h, so that the sample fiber was dispersed into a single fiber. Finally, the fibers were dripped with Herzberg stain [10], and they appear red-wine color [11] under the microscope in Figure 6b. In figure, the red marks were the cross stripes on the fiber, which formed nodes. This was characteristic of the most-used base fibers in ancient China (e.g., hemp, ramie, mulberry and paper mulberry). The lumen in the fiber could be clearly seen in the red label, which is typical of ramie fiber [11,25]. Ramie was most likely the source of the fiber in the mortar from the painting from the Taidong Tomb.

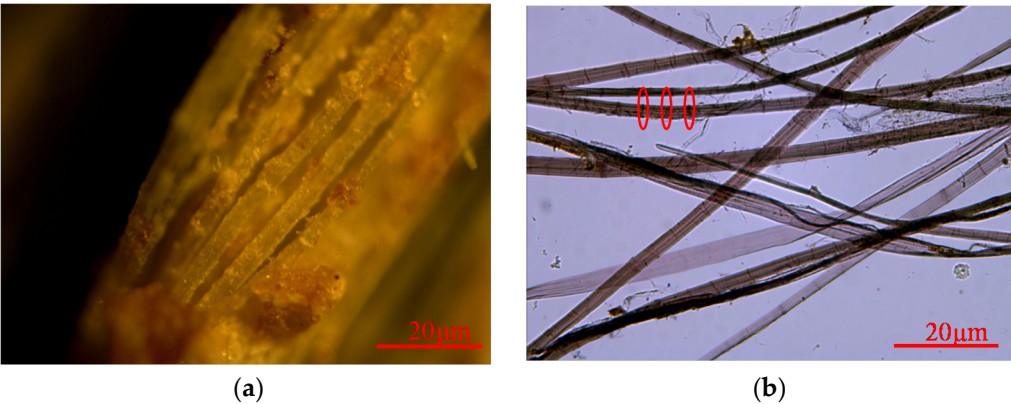

(**a**)            (**b**)

**Figure 6.** (**a**) Photo of untreated ramie fibers under a polarizing microscope at 20×; (**b**) photo of ramie fibers under a polarizing microscope at 20× and treated with Herzberg stain.

The X-ray diffraction results of the mortar shown in Figure 7 indicate that there were peaks at 2θ values of 21°, 26°, 36°, 39°, 41°, 42°, 45°, 50°, 55°, 60°, and 68°, respectively, coinciding with the diffraction peaks of the mineral gismondine ($CaAl_2Si_2O_8 \cdot 4H_2O$; JCPDS 20–0452). The peak at 28° was equal to the characteristic peak of albite phase ($Na(AlSi_3O_8)$, JCPDS 84–0752). The diffraction peaks at 30° and was equal to the characteristic peak of calcite phase ($CaCO_3$; JCPDS 43–0697), which may be produced by the lime water in the mortar absorbing carbon dioxide from the air. The diffraction peaks appeared at 22°, 24°, 31°, and 35°, corresponding to the graphitized coke produced by the carbonization of cross-linked tung oil. According to [9], the infrared spectrum of Figure 8 was similar to tung oil. Tung oil was a kind of dry vegetable oil, which originated in Asia. Tung oil was composed of

triglycerides, saturated fatty acids and unsaturated fatty acids. The high unsaturation of tung oil mainly comes from α-eleostearic acids [26]. The bands around 2921 and 2850 cm⁻¹ were caused by symmetric and asymmetric C–H stretching vibrations of methyl and methylene groups [27]. The peaks near 1459 and 759 cm⁻¹ were caused by asymmetric deformation of –CH₃ and rocking of CH₂, respectively. The weak peak at 1542 cm⁻¹ may be caused by C=O stretching under the conjugate action of C=C group generated by tung oil oxidation. Meanwhile, as tung oil ages, the vibration of C=O the ester group at the peak of 1742 cm⁻¹ in tung oil gradually disappeared and a new peak appeared at 1706 cm⁻¹, which may be caused by the stretching of C=O fatty acids [28]. In addition, the peaks at near 990 and 965 cm⁻¹, with the degradation of C–H wagging vibrations of conjugated double bonds, of trans, trans and cis, trans-configured [27], the peak at 965 cm⁻¹ almost disappeared [9].

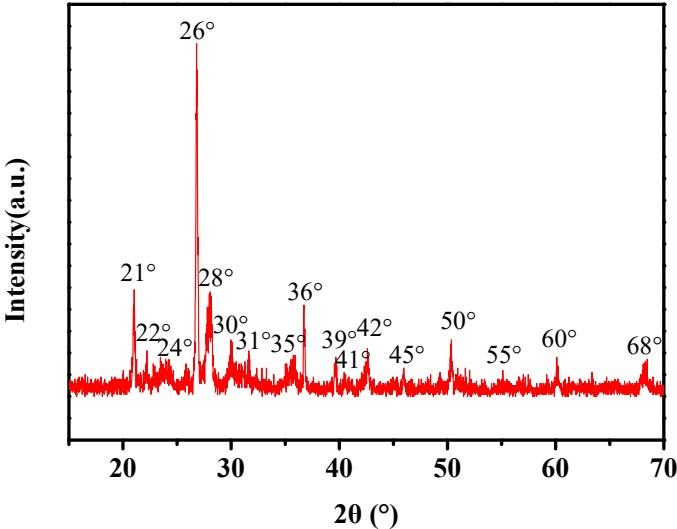

**Figure 7.** X-ray diffraction spectra of inorganic fillers in sample mortar.

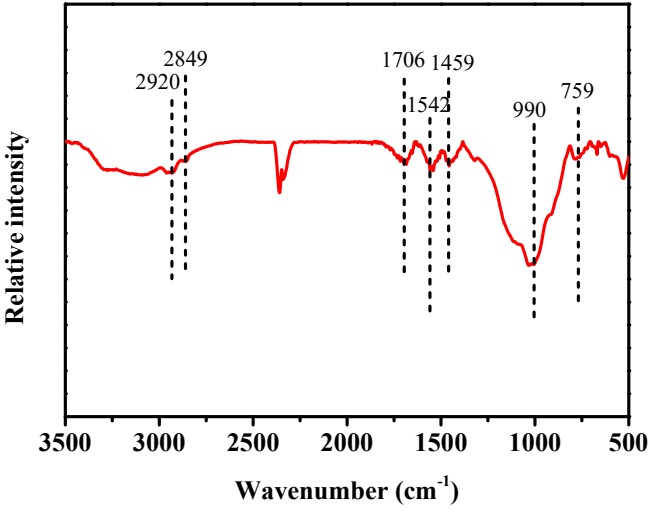

**Figure 8.** Fourier-transform infrared spectroscopy imaging of the binding in sample mortar.

Pyrolysis–gas chromatography–mass spectrometry analysis was used to further verify results,.Mortar of 50 µg was added to the thermal cracking instrument, and 3 µL of 20% tetramethylammonium hydroxide (TMAH) was added. After standing for 30 min, the sample was put into the autosampler for pyrolysis at 600 °C. After pyrolysis, the product was identified using gas chromatography–mass spectrometry.

Figure 9 is the total ion chromatogram of the mortar sample plus TMAH. The pyrolysis products are shown in Table 5 in which the 8 and 12 peaks correspond to the substance 2,3,6-Tri-O-methyl-d-glucopyranose and methyl alpha-D-mannopyranoside. This illustrates that the mortar layer also contained polysaccharides, which may come from flour.

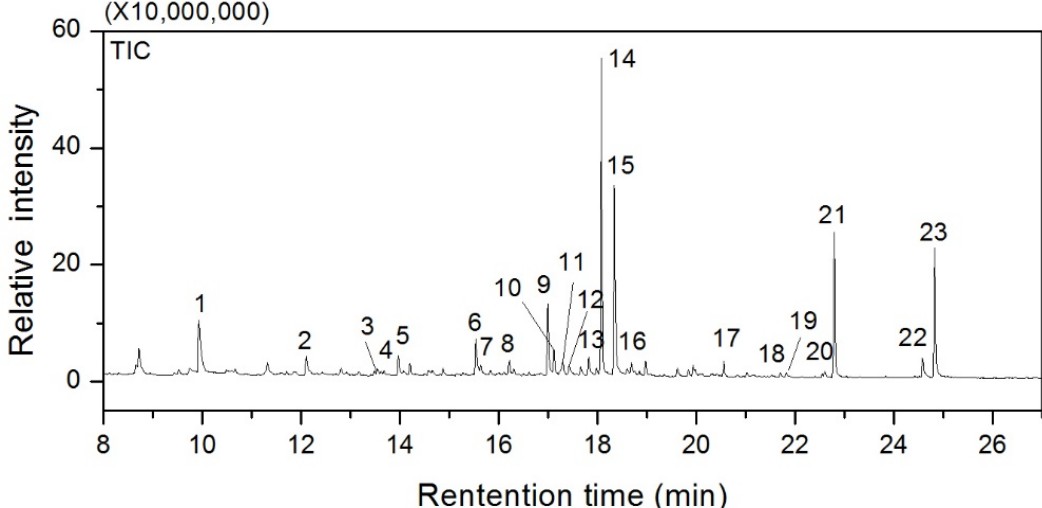

**Figure 9.** Total ion chromatography of mortar samples.

**Table 5.** A list of compounds identified by total ion chromatography of mortar obtained by pyrolysis–gas chromatography–mass spectrometry.

| Peak Number | Retention/Time (min) | Characteristic Components and Typical Segments in Mass Spectrometry | Area (%) |
|---|---|---|---|
| 1 | 9.92 | butanedioic acid, dimethyl ester | 3.38 |
| 2 | 12.10 | pentanedioic acid, dimethyl ester | 0.58 |
| 3 | 13.54 | butanoic acid, 4-(dimethylamino)-3-hydroxy- | 0.12 |
| 4 | 13.67 | nonanoic acid, methyl ester | 0.49 |
| 5 | 13.96 | hexanedioic acid, dimethyl ester | 0.21 |
| 6 | 15.53 | heptanedioic acid, dimethyl ester | 2.35 |
| 7 | 15.63 | 10-undecenoic acid, methyl ester | 0.16 |
| 8 | 16.20 | 2,3,6-tri-o-methyl-d-glucopyranose | 0.22 |
| 9 | 16.99 | octanedioic acid, dimethyl ester | 4.98 |
| 10 | 17.11 | dimethyl phthalate | 0.54 |
| 11 | 17.29 | hexanedioic acid, 3-methoxy-, dimethyl ester | 0.27 |
| 12 | 17.4 | methyl alpha-d-mannopyranoside | 0.21 |
| 13 | 17.81 | 1,4-benzenedicarboxylic acid, dimethyl ester | 0.24 |
| 14 | 18.07 | dodecanoic acid, methyl ester | 28.95 |
| 15 | 18.33 | nonanedioic acid (azelaic acid), dimethyl ester | 19.51 |
| 16 | 18.68 | octanedioic acid, 4-methoxy-, dimethyl ester | 0.23 |
| 17 | 20.55 | methyl tetradecanoate | 0.26 |
| 18 | 21.70 | pentadecanoic acid, methyl ester | 0.11 |
| 19 | 21.81 | 1,2,4-benzenetricarboxylic acid, trimethyl ester | 0.13 |
| 20 | 22.59 | 1,3,5-benzenetricarboxylic acid, trimethyl ester | 0.11 |
| 21 | 22.78 | palmitic acid, methyl ester | 18.49 |
| 22 | 24.57 | 9-octadecenoic acid, methyl ester | 0.55 |
| 23 | 24.81 | stearic acid, methyl ester | 17.91 |

The pyrolysis products contained mostly oxidation products of unsaturated fatty acids such as No. 1 peak—butanedioic acid, No. 2 peak—pentanedioic acid, No. 5 peak—hexanedioic acid, No. 6 peak—heptanedioic acid, No. 9 peak—octanedioic acid, No. 14 peak—dodecanoic acid and No. 15 peak—azelaic acid. The presence of dicarboxylic acid was due to the formation of autotrophs

by oxidation of polyunsaturated fatty acids in drying oils [29]. Among them, the content of azelaic acid was relatively high, which was the oxidation product of tung oil after heat treatment [10]. In addition, 9,10-dihydroxy-octadecanoic acid was not detected in pyrolysis products. Studies show that 9,10-dihydroxy-octadecanoic acid only appeared in hemp seed oil, poppy oil and wax oil, but not in tung oil [30]. To sum up, it was likely that the mortar contained drying oils (tung oil) and flour.

## 4. Conclusions

This work made a preliminary investigation of the materials used in architectural paintings from the Taidong Tomb in the Western Qing Tombs. Through the analysis of the damaged samples, three layers of structure were obtained from the cross sections of Xipei Hall and Long'en Hall, including a ramie layer, a mortar layer and a paint layer. The Fourier-transform infrared spectroscopy and pyrolysis–gas chromatography–mass spectrometry analysis showed that the mortar may contain tung oil and flour. Lime water and brick ash (gismondine, albite) were proven by using X-ray diffraction. The possibility of ramie fiber in mortar was studied by Herzberg stain. Finally, energy dispersive X-ray spectroscopy, polarized light microscopy, micro-Raman spectroscopy and X-ray diffraction were used to analyze the pigments for Long'en Hall, Xipei Hall and the ceiling of Minglou. The results were as follows: The green pigments were mainly atacamite and emerald green; the blue pigments were mainly azurite and ultramarine; the white pigments were lead-white and anglesite; the red pigments were vermilion; and the black pigments were carbon black. Emerald green and ultramarine were synthetic pigments, suggesting that the building was renovated sometime around the mid-to-late 19th century. Knowing the components of the paintings is crucial for their protection and any future restorations that may be needed. Taidong Tomb is a protected cultural legacy site; further archaeological research is necessary for the perseveration of Chinese culture.

**Author Contributions:** Conceptualization, Y.-H.L.; investigation, P.F.; data curation, P.F. and G.-L.T.; writing original draft preparation, J.-X.L.; writing review and editing, P.F. and J.L.; project administration, H.Y. All authors have read and agreed to the published version of the manuscript.

**Funding:** This research was sponsored by the Fundamental Research Funds for the Central Universities (2019TS002).

**Acknowledgments:** Thanks to Yang Hong from the Palace Museum for participating in the preliminary investigation and discussion of the Taidong Tomb and thanks to Yu-Hu Li for setting up the framework of the entire project. Thanks for the fund support (the Fundamental Research Funds for the Central Universities, 2019TS002).

**Conflicts of Interest:** The authors declare no conflict of interest.

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
