# Peer review of "Investigation of Ancient Architectural Painting from the Taidong Tomb in the Western Qing Tombs, Hebei, China"

_coatings, doi:10.3390/coatings10070688_

Round 1

Reviewer 1 Report

The paper “Investigation of Ancient Architectural Painting from the Taidong Tomb in the Western Qing Tombs, Hebei, China” provide a complete characterization of materials and manufacture techniques related to paintings of Taidong tomb. A combination of six analytical techniques was used for obtaining the identification of pigments and mortar layer. The investigation evidences a repainting of architecture structure in the mid-to-late 19th century.

The paper offers consistent results obtained with the complementary use of analytical techniques. These results could be useful for conservation and any future restorations of the site. I suggest the publication on Coating with minor revisions (see below):

  • the introduction concerning the importance of the site should be extended in order to emphasize the funerary architecture techniques, especially in relation to painting manufacturing. There are other similar sites? Few information about the painters or painting school could be an adding value to the manuscript.
  • Results and discussion: obtained results are compatible with other similar sites or painting school? In case of unicity, I suggest emphasizing it in the text.
  • Table 2 – Cross section micrograph of sample X3: the markers are not clear
  • Figure 3a: It is very difficult to distinguish the bands at 145, 180, 250, 284 and 766 cm-1because of the high noise in the spectrum. I suggest removing the labels and concentrating only to the main peaks at 403, 841, 1100, 1433, 1573 cm-1 (change also in the text at row 208)

Reviewer 2 Report

In this paper the authors have provided a multi-analytical characterization of pigments, mortars and fibers. I recommend this paper for publication in this journal. However, I would invite the authors to address the following changes in order to improve the quality and readability of the manuscript:

The English language requires an extensive editing by a native speaker. Several sentences lack a sense of fulfillment. Please make sure that the text is clear and complete!

Line 15. It's more correct to write "polarized light microscopy". Make this change also through the whole text.

In the "cross-section preparation" paragraph I would recommend to use the impersonal form and not the personal form. 

Line 118.  the authors should specify to what is referred the luminous efficiency percentage they are reporting. Is this value related to a lamp in the Raman system? Please specify.

Line 120. if you report the accumulation time you should indicate the unit of measurement associated to these accumulations. Is it 1-3 seconds? If in your instrument it is indicated simply the number of accumulations then you should write "accumulations" and not "accumulation time".

Which optical objectives have been used for data collection?

Line 128. It does not have any sense to report these measures with "better". What do the authors mean? Please, correct this part. "better than 0.16 cm-1; SNR: better than 55,000:1; Spatial resolution: better than 5 μm.

Line 148. I would report the 1200μm thickness as a 1,2mm. I would do the same for 2200μm (line 149) and 1000μm (line 150)

Line 157. Its "The element composition of the samples". Please, correct

In the caption of figure 3 I would indicate the name of the compound identified for each Raman spectrum. It would be also better for the reader if the name of the compound was reported inside the Raman spectrum.

Line 178. I would not say "caused by" but rather "attributed to" or associated to". Please, correct this along the text.

Line 192. What do the author mean as for "pdf card"? Please, clarify this.

Line 194. I suggest the authors to report at least a picture of the investigated pigment particles along the text or in the supplementary material. Please, provide this picture for each of the investigated pigment.

Line 209. You should add that "For azurite pigment, the bands found in the spectral interval 0-600 cm-1, are attributed to the Cu-O band vibration". Please check that the sentences and descriptions are complete in the whole manuscript!

Line 217 "Azurite and malachite are strongly associated minerals" This sentence is not clear, please, clarify this point.

Line 269. In this sentence it is not correct to talk about "nodes" rather "knots".

Line 321. please, correct this "Pyraphy-mass spectrometry"

Line 341. Please, rewrite the acknowledgements in an adequate and professional form.
